# Comprehensive Metabolite Profiling of Four Different Beans Fermented by *Aspergillus oryzae*

**DOI:** 10.3390/molecules27227917

**Published:** 2022-11-16

**Authors:** Yeon Hee Lee, Na-Rae Lee, Choong Hwan Lee

**Affiliations:** 1Department of Bioscience and Biotechnology, Konkuk University, Seoul 05029, Republic of Korea; 2Research Institute for Bioactive-Metabolome Network, Konkuk University, Seoul 05029, Republic of Korea

**Keywords:** beans, fermentation, *Aspergillus oryzae*, metabolite profiling, antioxidant activity

## Abstract

Fermented bean products are used worldwide; most of the products are made using only a few kinds of beans. However, the metabolite changes and contents in the beans generally used during fermentation are unrevealed. Therefore, we selected four different beans (soybean, *Glycine max*, GM; wild soybean, *Glycine soja*, GS; common bean, *Phaseolus vulgaris*, PV; and hyacinth bean, *Lablab purpureus*, LP) that are the most widely consumed and fermented with *Aspergillus oryzae*. Then, metabolome and multivariate statistical analysis were performed to figure out metabolite changes during fermentation. In the four beans, carbohydrates were decreased, but amino acids and fatty acids were increased in the four beans as they fermented. The relative amounts of amino acids were relatively abundant in fermented PV and LP as compared to other beans. In contrast, isoflavone aglycones (e.g., daidzein, glycitein, and genistein) and DDMP-conjugated soyasaponins (e.g., soyasaponins βa and γg) were increased in GM and GS during fermentation. Notably, these metabolite changes were more significant in GS than GM. In addition, the increase of antioxidant activity in fermented GS was significant compared to other beans. We expect our research provides a basis to extend choice for bean fermentation for consumers and food producers.

## 1. Introduction

Fermented beans have been used as an important nutrition source for proteins, lipids, vitamins, and phytochemicals for thousands of years [1,2,3]. Recently, fermented beans such as doenjang, ganjang, natto, doubanjiang, and tianmianjiang have been worldwide for flavors and health [4,5]. Since interest in health is growing, consumption of the fermented bean products is drastically increased nowadays [6,7]. It is well known that the fermented beans have antioxidant, anticarcinogenic, and hepatoprotective properties which are relevant to flavonoids and phenolic compounds [8].

Beans are commonly fermented by various microorganisms such as *Aspergillus* spp., *Rhizopus* spp., *Bacillus* spp., and *lactobacillus* spp. [1,9,10,11]. Among them, *Aspergillus oryzae* is widely used for fermentation in foods such as sake, vinegars, and fermented beans [10]. Since *A. oryzae* is highly harnessed in the food fermentation industry, the microorganism is well studied [1]. It is known that *A. oryzae* has high protease, amylase, and glucosidase activities, which are necessary for breaking down proteins and oligosaccharides in the raw food materials [9]. Moreover, the microorganism is substantially used for imperative secondary metabolite microbial cell factories. 

Most bean fermentations using *A. oryzae* are conducted with soybean (*Glycine max*, GM), and thus research on bean fermentation is mostly limited to GM. However, it is reported that other beans, apart from the GM, contain more bioactive substances than GM [12]. Therefore, in this study, we utilized four different beans fermented with *A. oryzae*. To compare the metabolite contents in the four fermented beans, we performed metabolite profiling based on gas chromatography time-of-flight mass spectrometry (GC-TOF-MS) and liquid chromatography-linear trap quadrupole-orbitrap-tandem mass spectrometry (UHPLC-LTQ-Orbitrap-MS/MS) analyses. Additionally, to evaluate the bioactivity changes of fermented beans, antioxidant activity assays were conducted. 

## 2. Results and Discussion

### 2.1. Metabolite Profiling of Four Different Beans before Fermentation

We selected four different beans, (1) soybean, (2) wild soybean (*Glycine Soja*, GS), (3) common bean (*Phaseolus vulgaris*, PV), and (4) hyacinth bean (*Lablab purpureus*, LP), widely used and having different properties (Figure 1A). It is reported that GM contains large amounts of proteins, carbohydrates, lipids, and secondary metabolites such as isoflavones, phytosterols, and soyasaponins [13]. GM is commonly used for cooking oils, fermented foods, and animal feeds. GS, the ancestor of GM, is a wild crop that has recently started to attract attention [14]. It is known that the general characteristics of GS are very similar to GM [15,16,17]. PV also has high protein, starch, and flavonoid contents, similar to the other beans. Interestingly, PV contains tannin, and anthocyanins, existing in the skin of the bean [18]. LP is a source of various nutrients such as proteins, tocopherols, minerals, vitamins, and phytochemicals, and is mainly used for medicine in East Asia (e.g., India, Bangladesh, Indonesia, and Korea) [19,20,21]. 

To compare the metabolite contents in the four beans, we performed the metabolite profiling and multivariate analyses using GC-TOF-MS and UHPLC-Orbitrap-MS/MS analyses. The principal component analysis (PCA) model obtained from the GC-TOF-MS analysis shows that GM and GS were clearly separated from PV and LP by PC 1 (Figure 1B). Intriguingly, a similar pattern was observed in the PCA score plot for the UHPLC-Orbitrap-MS/MS analysis result. It is noted that GM and GS in the PCA score plot of the UHPLC-Orbitrap-MS/MS analysis look almost same (Figure 1C). To investigate the different metabolites in the four beans, we profiled significantly discriminated metabolites based on the VIP value (>0.7) using partial least squares discriminant analysis (PLS-DA) and a *p*-value < 0.05. A total of 49 metabolites, including 18 amino acids, 11 fatty acids, 7 organic acids, and 7 carbohydrates, were tentatively identified based on the GC-TOF-MS data analysis (Appendix A). Additionally, according to the UHPLC-Orbitrap-MS/MS data analysis, we tentatively identified 33 metabolites including 6 flavonoids, 9 isoflavones, 6 soyasaponins, and 10 lipids.

To express the relative content of the significantly different metabolites in each bean, all metabolites were displayed on a heat scale (Figure 2A,B). Each column expresses fold changes calculated from the average peak area of each sample. In the heatmap analysis (Figure 2A,B), we could find similar patterns as for the PCA. GM and GS contain relatively higher amount of isoflavones and soyasaponins than PV and LP. Unexpectedly, amino acids were more abundant in PV and LP than the other beans. GM and GS have the most abundant carbohydrates and phospholipids among the four beans. Intriguingly, flavonoids, such as catechin and epicatechin, had the highest amounts in PV among the four beans (Figure 2B) [22]. 

### 2.2. Metabolite Changes of the Four Different Beans during Aspergillus-Oryzae-Mediated Fermentation

#### 2.2.1. Multivariate Statistical Analyses and Heatmap Analyses of the Four Beans during Fermentation

To evaluate metabolite changes during fermentation in the four different beans, we fermented steamed beans with *A. oryzae* for 3 days and harvested every day for further metabolome analysis. Subsequently, multivariate statistical analyses were conducted to discover significantly different metabolites during fermentation (Figure 3A,B).

Based on the PCA score plots, we could observe the progress of fermentation. Days 0, 1, 2, and 3 after fermentation were labeled as 0D, 1D, 2D, and 3D, respectively. According to the PCA plot of GC-TOF-MS analysis, we could find that 0D and 1D were very similar, which suggests that metabolites in all beans were negligibly changed until 1D (Figure 3A). After that, large metabolite changes during the fermentation were observed from 1D, indicating the fermentation might be drastically progressed from 1D. We could find consistent patterns in the heatmap analysis results (Figure 3C). There were large shifts in the contents of amino acids, fatty acids, and carbohydrates between 1D and 2D. In the four beans, the contents of amino acids and fatty acids were elevated during fermentation. The amino acids might be decomposed from the bean proteins by proteases during fermentation [4,23]. On the other hand, carbohydrates were decreased, which might be catalyzed by α-amylase from *A. oryzae* during fermentation [1,4,24]. Particularly, a comparatively larger amounts of carbohydrate were decreased in GM than in the other beans. Interestingly, at 3D, the amino acid contents in PV and LP were substantially higher than in GM and GS. LP showed the highest amino acid contents among the four beans at 3D (Figure 3A).

Based on the PCA using UHPLC-LTQ-Orbitrap-MS/MS analysis, we figured out very different patterns between the *Glycine* spp. and the others (Figure 3B). Unexpectedly, the secondary metabolite changes in *Glycine* spp. were less than for the other beans during fermentation. Unlike GM and GS, PV and LP showed greater changes, and the shifting patterns resemble the PCA plot for GC-TOF-MS analysis, which started changing from 1D. The PCA patterns are inconsistent with heatmap displayed in Figure 3D. The shifting patterns from 1D in PV and LP might be relevant with discriminant non-identified metabolites displayed in Appendix A. Surprisingly, LP contains a negligible amount of flavonoids, isoflavones, and soyasaponins as compared to the other beans. Moreover, in PV, flavonoids and a few soyasaponins were detected. Through the fermentation, the contents of isoflavone aglycones (e.g., daidzein, glycitein, and genistein) were increased whereas the relative contents of isoflavone glycosides (e.g., daidzin, glycitin, and genistin), and isoflavone acetyl glycosides (e.g., acetyldaidzin, acetylglycitin, and acetylgenistin) were decreased in GM and GS (Figure 3D). We guess that this is caused by β-glucosidase catalyzing conversion of isoflavone glycosides to isoflavone aglycones [25,26,27]. Moreover, in GM and GS, the relative amount of non-DDMP (2,3-dihydro-2,5-dihydroxy-6-methyl-4H-pyran-4-one)-conjugated soyasaponins (e.g., soyasaponins I, II, III, and IV) were decreased whereas the contents of DDMP-conjugated soyasaponins (e.g., soyasaponins βa and γg) were increased. During fermentation, most flavonoids were decreased in GS and PV as reported in previous reports [22]. However, quercetin 3-O-sambubioside, and apigenin were increased in GS. As a result, we figured out that the distinct metabolite composition in various beans led to different metabolite changing patterns even they were fermented with the same microorganism.

#### 2.2.2. Metabolite Pathway Analysis of the Four Beans Fermented with A. oryzae over Time

To compare metabolite changes of various beans before (0D) and after (3D) the fermentation using A. oryzae, we present the pathway map in Figure 4. Most amino acids (blue box in Figure 4) were elevated at 3D in the four beans. The relative contents of monosaccharides (e.g., glucose and galactose) and sugar alcohols (e.g., tyrosol, pinitol, and myo-inositol) in the four beans were decreased after fermentation (3D). The carbohydrates might be consumed by microorganisms for cell growing during fermentation [28]. However, maltose, which is a disaccharide, was increased. The relative contents of most lysophospholipids (orange box in Figure 4) were increased after fermentation. In the case of isoflavones (green box in Figure 4), the comparative contents of isoflavone glycosides (e.g., genistin, daidzin, and glycitin) and isoflavones with an acetyl glycoside group (e.g., acetylgenistin, acetyldaidzin, and acetylglycitin) were abundant in GM and GS before fermentation. At 3D, the relative contents of isoflavone aglycone (e.g., daidzein, genistein, and glycitein) were elevated in GM and GS, which might be influenced by β-glucosidase from microorganisms [25,29,30]. In the case of soyasaponins (red box in Figure 4), the soyasaponins I, II, III, and IV, non-DDMP-conjugated soyasaponins, were comparatively higher at 0D than 3D in GM and GS. On the other hand, the contents of DDMP-conjugated soyasaponins (e.g., soyasaponins βa and γg) were elevated in GM and GS at 3D. 

### 2.3. Antioxidant Activity, Total Flavonoid and Phenolic Contents

The antioxidant activity assay results show increasing patterns in GM and GS while the beans were fermented. On the other hand, in PV and LP, the antioxidant activities were negligibly changed. Notably, we observed a significant increase in the ABTS and FRAP assay results in GS as compared to the other beans (Figure 5). Especially, the antioxidant activity was greatly increased in GS at 2D. The increasing patterns were also observed in GS at 2D in the TFC and TPC assays. In addition, we observed an increase in the ABTS, FRAP, TFC, and TPC assays in GM at 3D. Therefore, we estimated that the increase in TFC and TPC is related to the enhancement of antioxidant activities. 

To evaluate the relevance between metabolite contents and the bioactivity of the four beans, correlation analysis was performed. Most amino acids showed negative correlations, but glutamine had a significant positive correlation with antioxidant activity (Appendix A) as similar as previous studies [31,32]. Most organic acids and carbohydrates showed a negative correlation with antioxidant activity. Flavonoids such as catechin, epicatechin, quercetin 3-O-sambubioside, and apigenin, represented a positive correlation. Among them, catechin, epicatechin, and quercetin 3-O-sambubioside showed significant positive correlation with antioxidant activity. Expectedly, most isoflavones showed a positive correlation as reported in previous study [33,34]. In particular, glycitin, glycitein, genistein, acetyldaidzin, acetlyglycitin, and acetylgenistin showed significant positive correlation with antioxidant activity. Moreover, soyasaponins, especially soyasaponins II, IV, βa, and γg, also showed a significant positive correlation. Unexpectedly, soyasaponins βa and γg, DDMP-conjugated soyasaponins, showed increasing patterns during fermentation. As a result, the antioxidant activity of fermented beans is highly influenced by the flavonoids and phenolic compounds such as genistein and soyasaponin βa, which were increased during A. oryzae fermentation. 

## 3. Materials and Methods

### 3.1. Chemicals and Reagents

HPLC-grade methanol, hexane, water, and acetonitrile were acquired from Fisher Scientific (Pittsburgh, PA, USA). All reagent-grade chemicals and standards were obtained from Sigma-Aldrich (St. Louis, MO, USA). Diethylene glycol and sodium carbonate were acquired from Junsei Chemical Co., Ltd. (Tokyo, Japan).

### 3.2. Microbial Cultures and Bean Fermentation

*Aspergillus oryzae* KACC (Korean Agricultural Culture Collection) 44967 was inoculated into Malt Extract Agar media and grown for 7 days at 30 °C in an incubator (Jeio tech, Daejeon, Korea). The spore suspension was prepared to have a spore count of 1 × 10^6^ spores/mL. 

The four beans were soaked in distilled water at room temperature (15–25 °C) for 12 h. After the soaked beans were autoclaved (Biofree, Seoul, Korea) at 120 °C for 15 min, 5 g of each bean was aliquoted into a 60 mm × 15 mm petri dish, and subsequently inoculated with 200 µL of the prepared *A. oryzae* suspension. Suspended fungi was inoculated into beans and fermented at 30 °C. The samples were harvested at 1 day (1D), 2 days (2D), and 3 days (3D). 

### 3.3. Sample Preparation

Each sample was dried using a freeze dryer (Operon, Gimpo, Korea) for 3 days and then ground with a mortar and pestle. The powdered samples were stored at *−*80 °C until metabolite extraction. 

Each sample of 100 mg was aliquoted into a 2 mL Eppendorf tube and 1 mL of 80% MeOH was added by dissolving 10 µL of 2-chloro-L-phenylalanine (1 mg/mL) with an internal standard. For sample homogenization, we utilized an MM400 mixer mill (Retsch, Haan, Germany) at 30 Hz for 10 min and then sonication treated for 5 min. The extract was centrifuged at 4 °C for 10 min at 21,000× *g* and filtered through a 0.22 µm polytetrafluoroethylene syringe filter (Chromdisc, Daegu, Korea). The filtered sample was completely dried utilizing a speed-vacuum concentrator (Biotron, Seoul, Korea). The dried sample was redissolved in 100% MeOH to a final concentration of 10,000 ppm (10 mg/mL). The samples were used for analysis and antioxidant activity measurements.

### 3.4. Gas Chromatography Time-of-Flight Mass Spectrometry Analysis

Each sample was subjected to two derivatization reactions according to the method that was described by Lee et al. [35]. For the GC-TOF-MS analysis, each sample extract (50 μL) was evaporated using a speed vacuum. Fifty microliters of pyridine-dissolved methoxyamine hydrochloride (20 mg/mL) were added to each dried sample and oximated at 30 °C for 90 min, reacted with 50 μL of MSTFA as a derivative, and incubated at 37 °C for 30 min.

The GC-TOF-MS analysis was performed with an Agilent 7890A GC system (Agilent Technologies, Palo Alto, CA, USA) that was equipped with an Agilent 7693 autosampler and Pegasus HT TOF-MS (Leco Corporation, St. Joseph, MI, USA). Helium was used as a carrier gas with a constant flow rate of 1.5 mL/min. An RTX-5MS column (30 m length × 0.25 mm i.d × 0.25 μm particle size; Restek Corp., St. Joseph, MI, USA) was used to separate the metabolites. Each derivatized sample was injected at 1 μL in a split of 15:1. The injector and ion source temperature were maintained at 250 °C. The column temperature was set to 75 °C for 2 min, increased to 300 °C at 15 °C/min, and finally maintained for 3 min. The detector voltage was 2152.5 V, and the mass was collected in the 50–700 *m*/*z* range. In addition, three analytical replicates were tested for each sample. The samples were analyzed in random blocks to reduce the impact of systematic errors, and then intermediate quality control samples were analyzed, consisting of pooled mixtures from each sample extract.

### 3.5. Liquid Chromatography-Linear Trap Quadrupole-Orbitrap-Tandem Mass Spectrometry Analysis

A UHPLC-LTQ-Orbitrap-MS/MS analysis was carried out using a UHPLC system with a Vanquish binary pump H system (Thermo Fisher Scientific, Waltham, MA, USA) array connected to an auto-sampler and column compartment. Chromatographic separation was performed using a Phenomenex KINETEX^®^ C18 column (100 × 2.1 mm, 1.7 μm particle size; Torrance, CA, USA), and the injection volume was 5 μL. The column temperature was retained at 40 °C, and the flow rate was 0.3 mL/min. The mobile phase consisted of 0.1% *v*/*v* formic acid in water (A) and 0.1% *v*/*v* formic acid in acetonitrile (B). The gradient parameters were set as follows: 5% B was retained for 1 min, followed by a linear increase to 100% B after 9 min, then it was sustained at 100% B for 1 min, and there was a gradual decrease to 5% B after 3 min. The total run time was 14 min. Also, the MS data were collected in the 100–1500 *m*/*z* range using an ion trap mass spectrometer (Thermo Fisher Scientific).

### 3.6. Data Processing and Multivariate Statistical Analysis

The raw data from the GC-TOF-MS and UHPLC-LTQ-Orbitrap-ESI-MS/MS analyses were converted to NetCDF (*.cdf). After that, the NetCDF (*.cdf) files were processed with the MetAlign software (http://www.metalign.nl (accessed on 13 October 2022)) to determine the data alignment by retention time (min) and peak mass (*m*/*z*) [36]. Multivariate statistical analysis was conducted using the SIMCA-P+ software (version 15.0.2), and PCA and partial least squares discriminant analysis (PLS-DA) modeling was carried out to compare different metabolites between the samples. The differentiated metabolites were then selected based on *p*-value (<0.05) and variable importance in the projection (VIP) value (>0.7). The selected metabolites were tentatively identified by comparing them with various analysis data such as molecular weights, retention time, formula, mass fragment patterns, the mass spectrum of the published references, standard compounds, the chemical dictionary version 7.2 (Chapman and Hall/CRC), an in-house library (offline database in the laboratory made by analyzing standards), and commercial databases such as the Human Metabolome Database (http://www.hmdb.ca/ (accessed on 21 May 2021)) and the National Institute of Standards and Technology Library (version 2.0, 2011, FairCom, Gaithersburg, MD, USA).

### 3.7. Antioxidant Activity and Total Flavonoid and Phenolic Contents

To determine the antioxidant activity of the fermented bean samples, 2,2′-azino-bis (3-ethylbenzothiazoline-6-sulfonic acid) diammonium salt (ABTS), ferric reducing antioxidant power (FRAP), total flavonoid content (TFC), and total phenolic content (TPC) assays were conducted in triplicate.

The ABTS and FRAP assays were performed using the method that was described by Lee et al. [37]. In brief, the ABTS stock solution, which was diluted with water to achieve a final absorbance of 0.7 ± 0.02 at 750 nm (180 μL), was added to 20 μL of each sample extract in a 96-well plate. After incubating in the dark at room temperature for 6 min. The absorbance was evaluated at 750 nm using a spectrophotometer. For the FRAP assay, a mixture of 10 mM 2,4,6-tripyridyl-S-triazine solution in 40 mM HCl (10:1:1, *v/v/v*), 300 mM acetate buffer (pH 3.6), and 20 mM iron (III) chloride prepared. Then, the sample (10 μL) was mixed with 300 μL of FRAP reagent and reacted for 6 min at room temperature. The absorbance was evaluated at 570 nm. The results of the ABTS and FRAP assays are expressed in Trolox equivalent antioxidant capacity (TEAC) concentration (mM) per milligram of the sample. The standard concentration curves ranged from 0.003906 mM to 1 mM TEAC in ABTS and from 0.003906 mM to 0.5 mM TEAC in FRAP.

For the TFC and TPC assays, a method described by Lee et al. [38] was followed. For the TFC assay, 20 µL of each sample was mixed with 180 µL of 90% diethylene glycol and 20 µL of 1 N NaOH in a 96-well plate. After reacting at room temperature for 60 min, the absorbance was evaluated at 405 nm. The TFC assay is expressed in naringin equivalent (NE) concentration (mM) per milligram of the sample. The standard concentration curves ranged from 3.90625 ppm to 500 ppm NE. For the TPC assay, 20 µL of each sample was incubated in a 96-well plate with 100 µL of 0.2 N Folin–Ciocalteu reagent at room temperature for 6 min. Then, 80 µL of 7.5% sodium carbonate (Na_2_CO_3_) solution was added and the plate incubated at room temperature for 60 min. Finally, the absorbance was measured at 750 nm. The standard concentration curves ranged from 0.390625 ppm to 100 ppm GE.

## 4. Conclusions

We examined the metabolite changes of four beans fermented with *A. oryzae*. We found out that the fermentation results are different depending on the bean (raw materials). We found similar fermentation patterns in most fermented beans. For example, carbohydrates were decreased and lipids were increased during fermentation. However, we found a discrepancy in the metabolite changing patterns for the four different beans. The relative content of amino acids were much higher in the PV and LP beans. In contrast, the relative content of isoflavone aglycones and soyasaponins were increased in GM and GS during fermentation. Antioxidant activity was increased when GS was fermented with *A. oryzae* as compared with GM. In GS, isoflavone glycoside has been converted to isoflavone aglycone by microorganisms, which is known to have a great influence on antioxidant activity. We hope that this study can provide valuable information to fermented food manufactures and consumers. In the future, to comprehend the metabolite changes with relevant enzymes, it will be necessary to examine the enzyme activities of the microorganism during the fermentation.

## Figures and Tables

**Figure 1 molecules-27-07917-f001:**
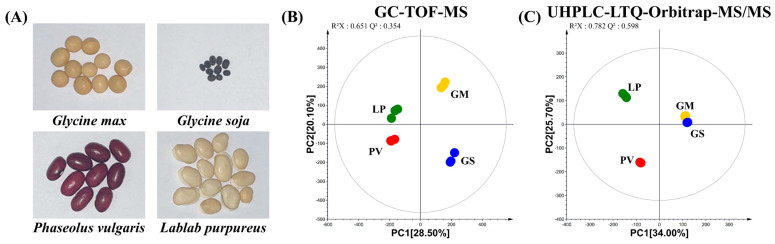
Photographs of the four types of beans (**A**), and a principal component analysis of the metabolites in the four beans before fermenting. The methods included gas chromatography time-of-flight mass spectrometry (GC-TOF-MS) (**B**) and liquid chromatography-linear trap quadrupole-orbitrap-tandem mass spectrometry (UHPLC-LTQ-Orbitrap-MS/MS) (**C**). (
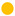
 (GM): *G. max*, 
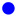
 (GS): *G. soja*, 
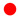
 (PV): *P. vulgaris*, 
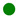
 (LP): *L. purpureus*).

**Figure 2 molecules-27-07917-f002:**
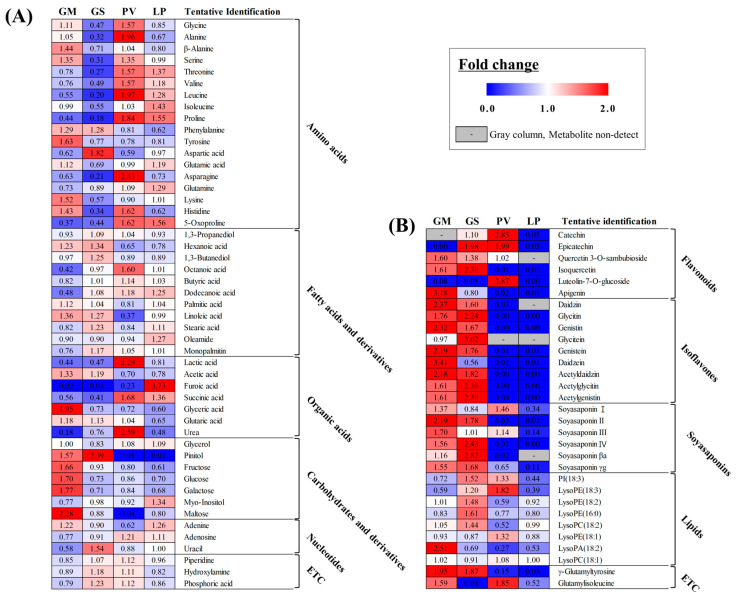
Heatmap analysis of four beans based on GC-TOF-MS (**A**) and UHPLC-LTQ-Orbitrap-MS/MS (**B**) analyses. The heatmap represents the relative content of significantly different metabolites determined by the partial least squares discriminant analysis (PLS-DA) model (VIP > 0.7, *p*-value < 0.05). The colored blocks represent the fold changes (blue to red) normalized by the average of all values for each metabolite. The colors indicate the relative abundances of each metabolite. Blue color: lower content relative to the mean, red color: higher content relative to the mean. GM: *G. max*, GS: *G. soja*, PV: *P. vulgaris*, LP: *L. purpureus*.

**Figure 3 molecules-27-07917-f003:**
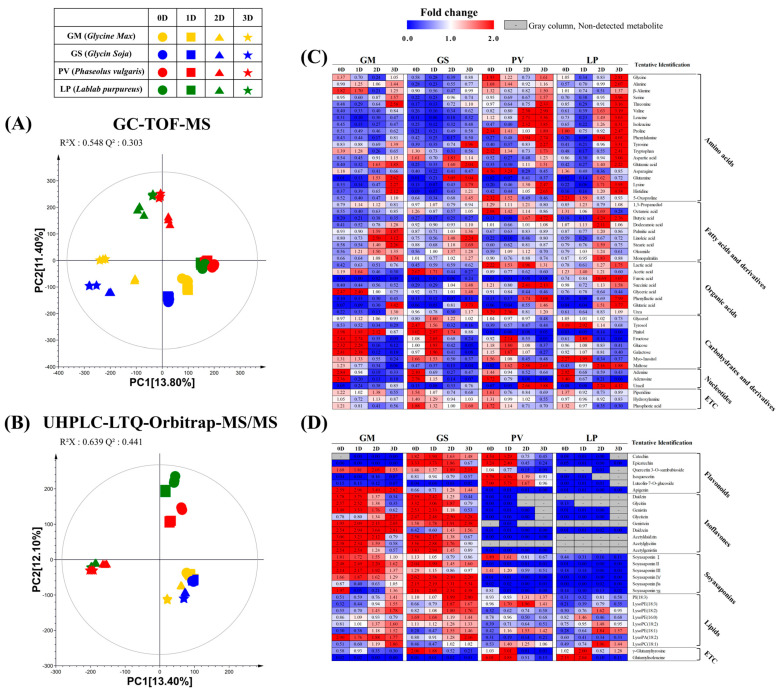
PCA analyses and heatmap analyses of the metabolites in the four beans that were fermented with *A. oryzae*. The methods included GC-TOF-MS (**A**,**C**) and UHPLC-LTQ-Orbitrap-MS/MS (**B**,**D**). The heatmap represents the relative content of significantly different metabolites determined by the partial least squares discriminant analysis (PLS-DA) model (VIP > 0.7, *p*-value < 0.05). The colored blocks represent the fold changes (blue to red) normalized by the average of all values for each metabolite. The colors indicate the relative abundances of each metabolite. Blue color: lower content relative to the mean, red color: higher content relative to the mean. GM: *G. max*, GS: *G. soja*, PV: *P. vulgaris*, LP: *L. purpureus*, 0D: 0 day of fermentation, 1D: 1 day after fermentation, 2D: 2 days after fermentation, 3D: 3 days after fermentation.

**Figure 4 molecules-27-07917-f004:**
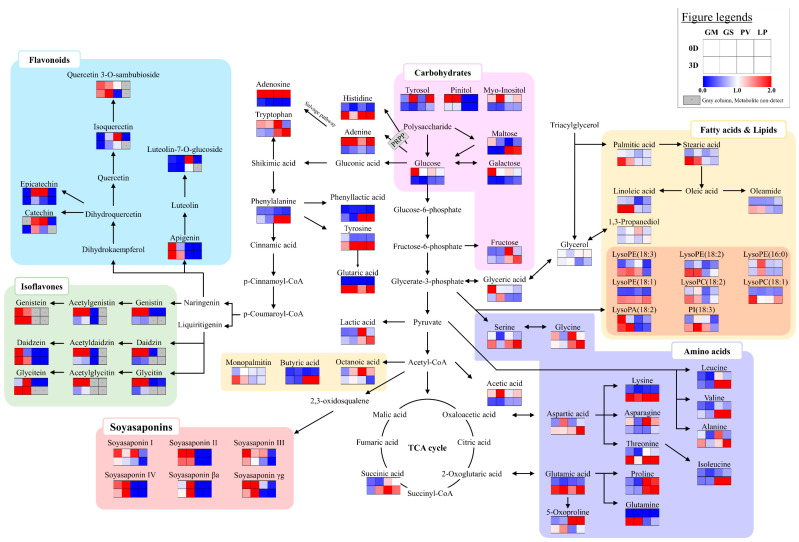
The pathway and heatmap analysis for the metabolites in the four beans that were fermented with *A. oryzae* 0 d and 3 d after fermentation. The metabolite pathways were adopted from the KEGG database. The colors symbolize relative abundance. GM: *G. max*, GS: *G. soja*, PV: *P. vulgaris*, LP: *L. purpureus*, 0D: 0 day of fermentation, 3D: 3 days after fermentation.

**Figure 5 molecules-27-07917-f005:**
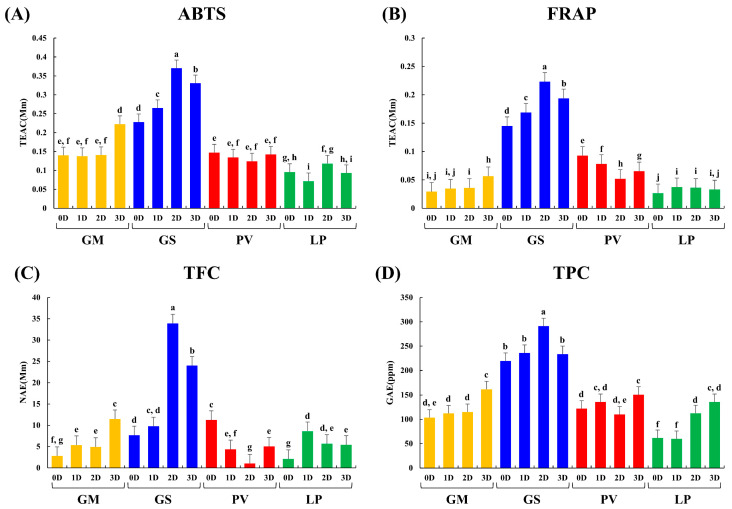
The antioxidant activity test, total flavonoids, and phenolic contents. (**A**) 2,2′-azino-bis (3-ethylbenzothiazoline-6-sulfonic acid) diammonium salt [ABTS], (**B**) ferric reducing antioxidant power [FRAP], (**C**) total flavonoid contents [TFC], and (**D**) total phenolic contents [TPC]. The significance was determined using Duncan’s multiple range test, and different letters indicate significant differences (*p*-value < 0.05). 
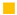
 (GM): *G. max*, 
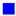
 (GS): *G. soja*, 
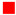
 (PV): *P. vulgaris*, 
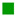
 (LP): *L. purpure*us, 0D: 0 day of fermentation, 1D: 1 day after fermentation, 2D: 2 days after fermentation, 3D: 3 days after fermentation.

## Data Availability

Not applicable.

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
