# Peer review of "Comprehensive Metabolite Profiling of Four Different Beans Fermented by Aspergillus oryzae"

_molecules, 2022, doi:10.3390/molecules27227917_

Round 1
Reviewer 1 Report
The manuscript is interesting and novel, it is necessary to attend to the following comments:
Line 27: the paragraph margin format is smaller than that used in other paragraphs
Line 35-41: the reference or references to the text of these lines is not shown
Line 68: Figure 1B?
Line 82,92,93,140,143,164,166,191,192: It could abbreviate the first word of the scientific name, since it was previously mentioned at the beginning of the paragraph
Line 96,145: According to the authors' guide, the text format for the titles of the Subsubsections should not appear in italic text format.
Line 202: could include information on the model, brand and country of the equipment used to incubate
Line 205: could include information on the model, brand and country of the equipment used (Autoclave)
Line 210: sometimes the word day is abbreviated by d, but sometimes, it is not abbreviated. Check
Line 213: ml or mL like in line 203
Line 246: um or µm
Line 273: modify….meaning? (ABTS),…..
Line 293,297: insert space…number ppm…
Line 339: according to the author guide, the title must appear in lowercase text format except for the first letter of the first word. Review through this section
Could you include in the study the determination of the inhibition of free radicals - DPPH?
Author Response
Reviewer 1
Dear reviewers,
We are grateful for the effort that reviewer provided valuable feedback on our manuscript. We have modified manuscript based on reviewer’s comments.
[Reviewer’s comment]
Line 27: the paragraph margin format is smaller than that used in other paragraphs
[Reply to the comment]
Based on reviewer’s comment, we modified the paragraph margin to match the other paragraphs.
[Reviewer’s comment]
Line 35-41: the reference or references to the text of these lines is not shown
[Reply to the comment]
Thank you for the valuable comments. We added references in the manuscript according to reviewer’s suggestion.
[Reviewer’s comment]
Line 68: Figure 1B?
[Reply to the comment]
Thank you for letting us know the typo. We modified it to Figure 1B.
[Reviewer’s comment]
Line 82,92,93,140,143,164,166,191,192: It could abbreviate the first word of the scientific name, since it was previously mentioned at the beginning of the paragraph
[Reply to the comment]
We thank the reviewer for the valuable suggestion. We changed it to the abbreviated nomenclature.
[Reviewer’s comment]
Line 96,145: According to the authors' guide, the text format for the titles of the Subsubsections should not appear in italic text format.
[Reply to the comment]
We thank the reviewer for the valuable suggestion. The subsubsections were changed to the desired format without italics in accordance to the authors’ guide and reviewer's comments.
[Reviewer’s comment]
Line 202: could include information on the model, brand and country of the equipment used to incubate
[Reply to the comment]
We thank the reviewer to bring it to notice. We added the information regarding the model, manufacturer, and country of the incubator used.
[Reviewer’s comment]
Line 205: could include information on the model, brand and country of the equipment used (Autoclave)
[Reply to the comment]
We thank for the suggestion. We added the information of autoclave used.
[Reviewer’s comment]
Line 210: sometimes the word day is abbreviated by d, but sometimes, it is not abbreviated. Check
[Reply to the comment]
We thank the reviewer for bringing the error to our attention. We changed ‘d’ to ‘days’ to avoid any confusion.
[Reviewer’s comment]
Line 213: ml or mL like in line 203
[Reply to the comment]
We thank the reviewer for bring it to notice. We changed ‘ml’ to ‘mL’ for uniformity with other parts.
[Reviewer’s comment]
Line 246: um or µm
[Reply to the comment]
We thank the reviewer for bring it to notice. We rectified ‘um’ to ‘µm’.
[Reviewer’s comment]
Line 273: modify….meaning? (ABTS),…..
[Reply to the comment]
We thank the reviewer to bring it to notice. We added full name of ABTS as follows:
2,2′-azino-bis (3-ethylbenzothiazoline-6-sulfonic acid) diammonium salt
[Reviewer’s comment]
Line 293,297: insert space…number ppm…
[Reply to the comment]
We thank the reviewer for the comment. We inserted a space in front of ppm.
[Reviewer’s comment]
Line 339: according to the author guide, the title must appear in lowercase text format except for the first letter of the first word. Review through this section
[Reply to the comment]
We thank the reviewer for the suggestion. We changed the reference section according to reviewer’s suggestion.
[Reviewer’s comment]
Could you include in the study the determination of the inhibition of free radicals - DPPH?
[Reply to the comment]
We thank the reviewer for the valuable suggestion. Due to time and resource limits, we were unable to perform the DPPH assay in this study. According to reviewer’s suggestion, we are seriously considering to include DPPH free radical inhibition assay in later studies.
Reviewer 2 Report
Methodology:
please provide detail of what process are performed in the heatmap analysis and explain the data meaning in the table of heatmap analysis in Figure 2A&B. If this means the content of each amino acid in each materials, please provide the standard deviation for better comparison. Similar indication are needed in Figure 3C.
Line 85: please explain or provide reference of why PV and LP has higher amino acid content.
Line 171: please add explanations on why the antioxidant activity increased at 2d fermentation in GS. This pattern is consistent for all four tests and different with other beans. This part worthy more exploration and discussion.
Author Response
Reviewer 2
Dear reviewers,
We thank the reviewer for evaluating our study and giving valuable comments. We tried to improve our manuscript based on the reviewer's comments.
[Reviewer’s comment]
please provide detail of what process are performed in the heatmap analysis and explain the data meaning in the table of heatmap analysis in Figure 2A&B. If this means the content of each amino acid in each materials, please provide the standard deviation for better comparison. Similar indication are needed in Figure 3C.
[Reply to the comment]
We thank the reviewer for the valuable comment. Based on the reviewer’s comment, we added the sentence in the caption to explain the meaning of heatmap as follows:
“The colored blocks represent the fold changes (blue to red) normalized by the average of all values for each metabolite. The colors indicate the relative abundance of each metabolite. Blue color : lower content relative to the mean, red color : higher content relative to the mean,”
Also, we added heatmap of standard deviations of Figure 2A, B, and 3C, 3D in Supplementary information Figure S5 and S6, respectively.
[Reviewer’s comment]
Line 85: please explain or provide reference of why PV and LP has higher amino acid content.
[Reply to the comment]
We appreciate the reviewer's comment. Although there are few papers comparing the amino acid content of various beans, we could not find studies that compared amino acid contents of PV and/or LP with GM and GS. We think that this result is the first to be revealed in this study.
[Reviewer’s comment]
Line 171: please add explanations on why the antioxidant activity increased at 2d fermentation in GS. This pattern is consistent for all four tests and different with other beans. This part worthy more exploration and discussion.
[Reply to the comment]
We thank the reviewer for the valuable suggestion. According to the reviewer’s comment, we added explanation and rectified conclusion in this subsection as follows:
“The antioxidant activity assay results show increasing patterns in GM and GS while the beans were fermented. On the other hand, in PV and LP, the antioxidant activities were negligibly changed. Notably, we observed a significant increase in ABTS and FRAP assay results in GS as compared to other beans (Figure 5). Especially, the antioxidant activity was greatly increased in GS at 2D. The increasing patterns were also observed in GS at 2D in TFC and TPC assays. In addition, we observed increase in all ABTS, FRAP, TFC, and TPC assays in GM at 3D. Therefore, we guessed that the increment of TFC and TPC is related to enhancement of antioxidant activities.”
“As a result, the antioxidant activity of fermented beans is highly influenced by the flavonoids and phenolic compounds such as genistein and soyasaponin βa, which were increased during A. oryzae fermentation.”